# Structural basis for inhibition of erythrocyte invasion by antibodies to *Plasmodium falciparum* protein CyRPA

Lin Chen[1,2][†], Yibin Xu[1,2][†], Wilson Wong[1,2], Jennifer K Thompson[1], Julie Healer[1], Ethan D Goddard-Borger[1,2], Michael C Lawrence[1,2], Alan F Cowman[1,2]*

[1]The Walter and Eliza Hall Institute of Medical Research, Melbourne, Australia; [2]Department of Medical Biology, University of Melbourne, Melbourne, Australia

**Abstract** *Plasmodium falciparum* causes malaria in humans with over 450,000 deaths annually. The asexual blood stage involves invasion of erythrocytes by merozoites, in which they grow and divide to release daughter merozoites, which in turn invade new erythrocytes perpetuating the cycle responsible for malaria. A key step in merozoite invasion is the essential binding of PfRh5/CyRPA/PfRipr complex to basigin, a step linked to the formation of a pore between merozoites and erythrocytes. We show CyRPA interacts directly with PfRh5. An invasion inhibitory monoclonal antibody to CyRPA blocks binding of CyRPA to PfRh5 and complex formation thus illuminating the molecular mechanism for inhibition of parasite growth. We determined the crystal structures of CyRPA alone and in complex with an antibody Fab fragment. CyRPA has a six-bladed $\beta$-propeller fold, and we identify the region that interacts with PfRh5. This functionally conserved epitope is a potential target for vaccines against *P. falciparum*.

*For correspondence: cowman@wehi.edu.au

[†]These authors contributed equally to this work

**Competing interests:** The authors declare that no competing interests exist.

## Introduction

The most severe form of malaria in humans is caused by *P. falciparum* with approximately 214 million cases and over 450,000 deaths each year occurring mostly in subtropical and tropical regions of the world (*Who, 2015*). Infection of humans occurs during blood feeding by a female Anopheles mosquito. The injected parasites migrate to the liver, and after development in hepatocytes liver merozoites are released that quickly invade erythrocytes in the circulating blood. The parasites grow and divide to produce 16 to 32 new daughter merozoites. Following egress from the host cell, these daughter cells invade new erythrocytes perpetuating the asexual blood stage life cycle that is responsible for the symptoms of malaria.

Invasion of human erythrocytes by *P. falciparum* merozoites involves multiple interactions of ligands with host receptors in a complex multistep process that ultimately ends with the internalization of the parasite (reviewed in [*Cowman and Crabb, 2006*]). The initial interaction of the parasite with the erythrocyte membrane is driven by low affinity interactions involving surface proteins that facilitate apical reorientation. This is followed by high-affinity binding of specific host receptors to the erythrocyte binding-like (EBL) and reticulocyte binding-like homologues (PfRh or PfRBP) ligand families to specific host receptors (reviewed in (*Cowman and Crabb, 2006*). The PfRh ligands are large proteins, released onto the surface of the merozoite and required for activation of downstream invasion events (*Rayner et al., 2000*, *2001*; *Triglia et al., 2001*; *Tham et al., 2015*, *2010*). PfRh5 is a disparate member of the PfRh family of ligands because, unlike other members of this protein family, it is small and lacks a transmembrane domain (*Hayton et al., 2008*; *Baum et al., 2009*). PfRh5 binds to basigin on the erythrocyte surface (*Crosnier et al., 2011*). The crystal

structures of PfRh5 alone (*Chen et al., 2014*) and in complex with its receptor basigin (*Wright et al., 2014*) have been determined and the Rh domain shown to exhibit a novel fold.

PfRh5 forms a complex with cysteine-rich protective antigen (CyRPA) and *P. falciparum* Rh5 interacting protein (PfRipr) (*Chen et al., 2011a*; *Reddy et al., 2015*; *Volz et al., 2016*). The function of PfRh5 is essential and blocking of its interaction with basigin using either soluble basigin or specific antibodies inhibits merozoite invasion (*Volz et al., 2016*; *Weiss et al., 2015*). Furthermore, *P. falciparum* merozoites in which the *cyrpa* or *pfripr* genes have been conditionally disrupted also cannot invade human erythrocytes and this process is blocked at the same point as observed when PfRh5 function is inhibited (*Volz et al., 2016*). The function of the PfRh5/CyRPA/PfRipr complex has been associated with the formation of a discontinuity or pore between the merozoite and the erythrocyte that allows movement of $Ca^{2+}$ into the host cell. It has also been hypothesized that this protein complex may be directly or indirectly involved in transfer of proteins into the host cell (*Volz et al., 2016*; *Weiss et al., 2015*). Regardless of the specific mechanisms at play, the PfRh5/CyRPA/PfRipr complex plays a pivotal role in the sequential molecular events leading to merozoite invasion of erythrocytes.

CyRPA and PfRipr are localized in the micronemes whereas PfRh5 is present at the neck of the rhoptries and these proteins are released onto the surface during merozoite invasion (*Volz et al., 2016*). Super-resolution microscopy has shown that the tripartite complex forms only at the interface between the invading parasite membrane and the erythrocyte membrane, with pools of PfRh5, CyRPA and PfRipr spread over the surface of the merozoite (*Volz et al., 2016*). The PfRh5/CyRPA/PfRipr complex is tightly associated with the membrane, and previous evidence suggested that CyRPA has a glycophosphatidylinositol (GPI) membrane anchor and is responsible for the association of the complex with the merozoite membrane (*Reddy et al., 2015*). However, it was subsequently shown that CyRPA does not have a GPI anchor, and, as none of these proteins have a transmembrane domain, it remains unclear how they associate with the parasite's plasma membrane (*Volz et al., 2016*).

Antibodies to PfRh5, PfRipr and CyRPA inhibit merozoite invasion and these proteins are important blood stage malaria vaccine candidates (*Chen et al., 2011a*; *Douglas et al., 2011*; *Williams et al., 2012*; *Patel et al., 2013*; *Reddy et al., 2014*). CyRPA-specific monoclonal antibodies inhibit growth of *P. falciparum* in a NOD-scid IL2Ry[null] mice engrafted with human erythrocytes (*Dreyer et al., 2012*). Additionally, a PfRh5-based vaccine has been shown to be efficacious in *P. falciparum* challenged aotus monkeys (*Douglas et al., 2015*).

To understand the molecular basis for blocking of CyRPA function by antibodies, we have co-crystallized this protein with a Fab fragment obtained from an invasion inhibitory monoclonal antibody. We have determined the crystal structure of this complex using diffraction data to 2.44 Å resolution and shown that CyRPA has a six-bladed $\beta$-propeller fold. We have also determined the structure of the uncomplexed CyRPA using data to 3.09 Å resolution. Together, these structures have enabled elucidation of the structural basis for antibody-mediated inhibition of CyRPA-enabled *P. falciparum* merozoite invasion.

## Results

### Recombinant expression of functional CyRPA

Full-length mature CyRPA (residues 30–362 of the translated protein product) was expressed in insect cells and purified to homogeneity as evidenced by size-exclusion chromatography (SEC) and SDS-PAGE analysis (*Figure 1A*). Whilst CyRPA has been shown to form a co-complex with PfRh5 and PfRipr (*Reddy et al., 2015*; *Volz et al., 2016*), it is not known with which of these two proteins it associates. To determine if the recombinant CyRPA can bind to PfRh5 (*Chen et al., 2014*), we combined the two proteins and showed by SEC they are capable of forming a stable 1:1 complex. In these experiments, PfRh5 was incubated with excess CyRPA and the stable CyRPA/PfRh5 complex eluted ahead of free CyRPA (*Figure 1B*). CyRPA/PfRh5 complex formation was also confirmed by incubation of excess CyRPA with FLAG-tagged PfRh5 followed by immunoprecipitation with anti-FLAG antibodies. In these experiments, CyRPA co-precipitated with PfRh5 (*Figure 1C*). Together with the SEC result, this finding suggests an association between CyRPA and PfRh5 within the PfRh5/CyRPA/PfRipr complex during merozoite invasion. These experiments also indicate that our recombinant CyRPA is correctly folded and also functionally competent.

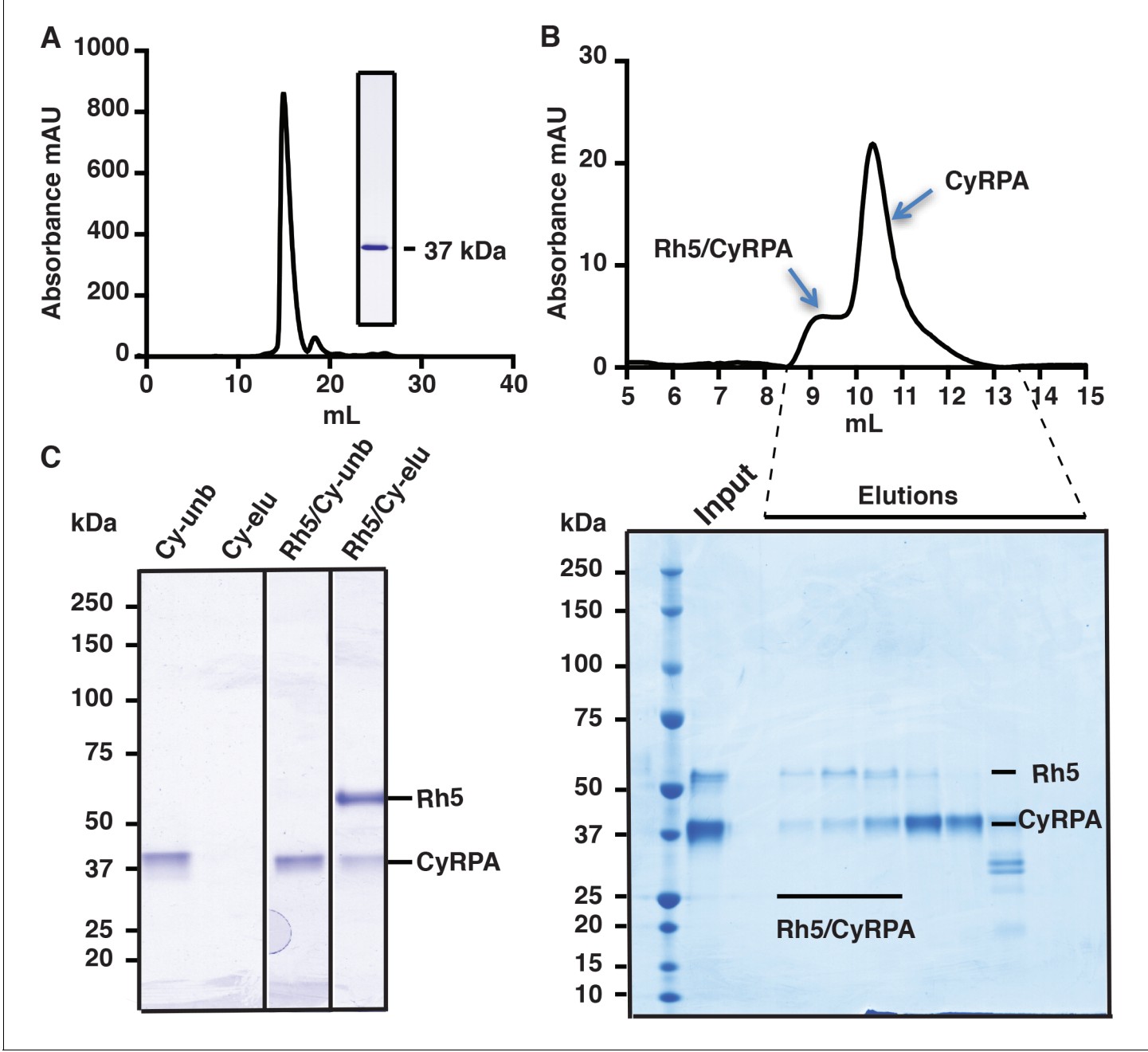

**Figure 1.** Production of functional recombinant CyRPA. (**A**) Purified recombinant CyRPA was analysed by size exclusion chromatography and SDS-PAGE. (**B**) Formation of the PfRh5/CyRPA complex was monitored by size exclusion chromatography, CyRPA formed a complex with Rh5 and the complex was eluted in the peak labelled as Rh5/CyRPA. The chromatographic profiles are shown (top panel) and fractions eluted from the column (lower panel) were analysed by SDS-PAGE. (**C**) Immunoprecipitation of FLAG-PfRh5 after incubation with CyRPA. Cy-unb, CyRPA alone; Cy-eluted, elution from anti-FLAG antibody beads of CyRPA alone; Rh5/Cy-unb, FLAG-PfRh5 and excess CyRPA incubated and unbound protein analysed; Rh5/Cy-elu, FLAG-PfRh5 and CyRPA eluted from anti-FLAG antibody beads.

## CyRPA-specific invasion inhibitory monoclonal antibodies block interaction of CyRPA with PfRh5

Monoclonal antibodies were raised to recombinant CyRPA and their ability to inhibit *P. falciparum* growth was determined. Three monoclonal antibodies (5B12, 3D1 and 8A7) were identified that inhibited growth to approximately 70% at 2 mg/ml, whereas others (7A6 and 8B9) showed

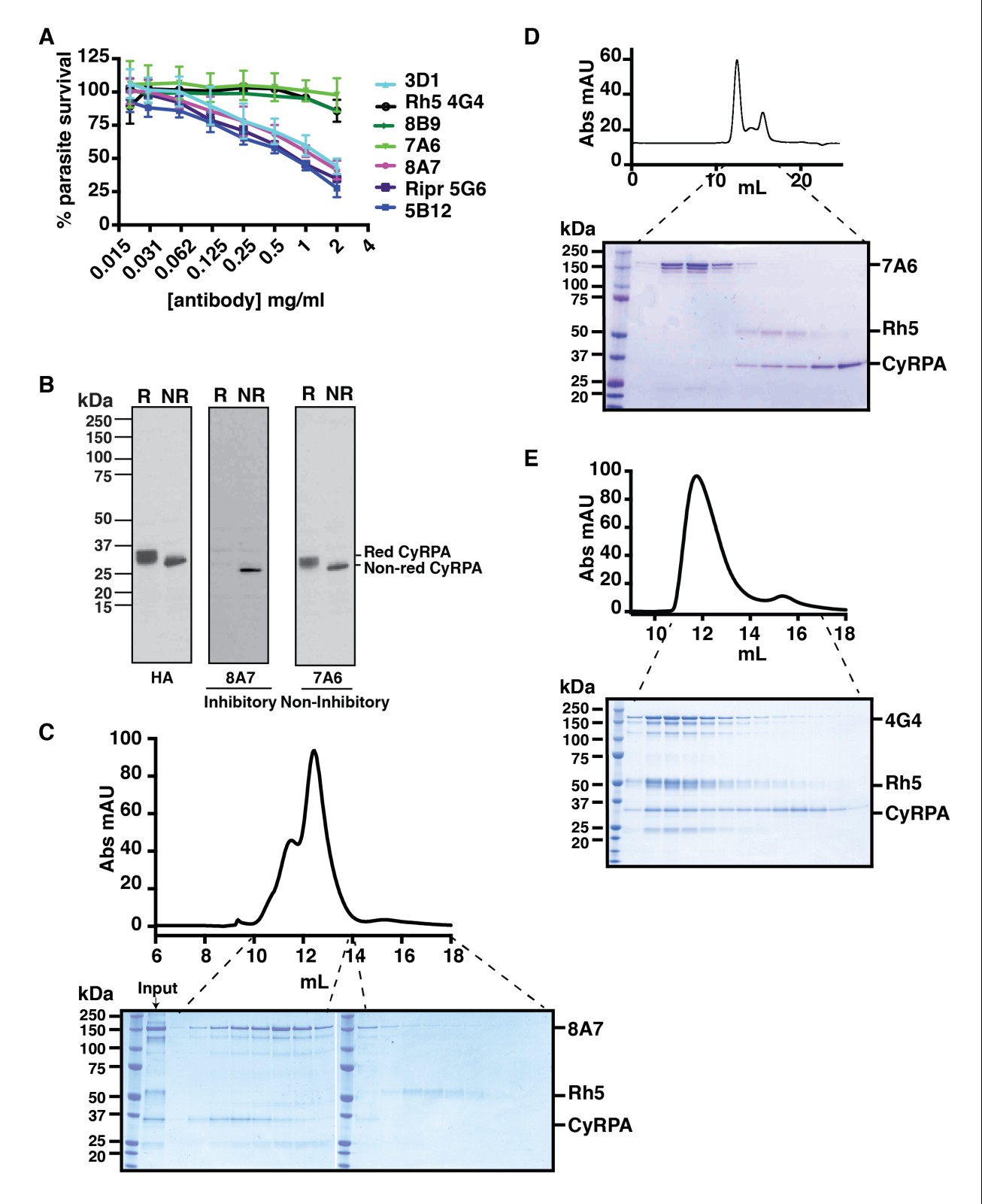

**Figure 2.** Anti-CyRPA monoclonal antibodies inhibit *P. falciparum* growth and interaction of PfRh5. (**A**) In vitro growth inhibition (GIA) assays were performed to assess the abilities of the monoclonal antibodies raised against recombinant CyRPA to block *P. falciparum* parasite growth in human erythrocytes. Assays were performed twice in triplicate and error bars denote SD of the mean of 6 values. Anti-Rh5 mAb 4G4 (not inhibitory) and anti-Ripr mAb 5G6 (inhibitory) are included as GIA controls. (**B**) Immunoblot of inhibitory (8A7) and non-inhibitory (7A6) monoclonal antibodies against

*Figure 2 continued on next page*

*Figure 2 continued*

proteins from CyRPA-HA tagged transgenic *P. falciparum* schizonts in reduced (R) and non-reduced (NR) condition. The *P. falciparum* parasites expressed haemagglutinin-tagged CyRPA protein. (C) Monoclonal antibody 8A7 blocked binding of PfRh5 to CyRPA. PfRh5 and CyRPA were incubated together with 8A7 and the complex formation monitored by size exclusion chromatography and SDS-PAGE analysis under non-reducing conditions. Monoclonal antibody 8A7 bound to CyRPA, preventing the PfRh5/CyRPA complex formation. CyRPA was co-eluted with 8A7 in the earlier fractions and the free Rh5 eluted in the later fractions. (D) Monoclonal antibody 7A6 did not inhibit the formation of the PfRh5/CyRPA complex as monitored by size exclusion chromatography and SDS-PAGE analysis. This antibody did not bind to native CyRPA indicating the linear epitope is not surface exposed. (E) Non-inhibitory anti-PfRh5 monoclonal antibody 4G4 did not significantly inhibit the formation of the PfRh5/CyRPA complex. The PfRh5/CyRPA complex bound to antibody 4G4 and was eluted together with 4G4 as a tri-molecule complex in the earlier fractions.

essentially no inhibition (*Figure 2A*). Immunoblot analysis of CyRPA with the monoclonal antibodies showed those that blocked parasite growth reacted with a conformational epitope as they bound to non-reduced protein but not to reduced protein (*Figure 2B*). In contrast, the antibodies that did not inhibit parasite growth reacted with both reduced and non-reduced CyRPA, consistent with them binding to a linear epitope. The three inhibitory anti-CyRPA monoclonal antibodies (5B12, 3D1 and 8A7) have a similar potency as the inhibitory anti-Ripr monoclonal antibody 5G6 in the growth inhibition assays (*Figure 2A*).

To determine the growth-inhibitory mechanism of the monoclonal antibody 8A7, we performed experiments to examine if the antibody blocked the interaction of CyRPA with PfRh5 (*Figure 2C*). Both proteins were incubated together in the presence of either inhibitory monoclonal antibody 8A7 or non-inhibitory monoclonal antibody 7A6. The mixtures were analysed by SEC and fractions visualized on SDS-PAGE and stained with Coomassie Blue R250 (*Figure 2C and D*). In the presence of inhibitory monoclonal antibody 8A7, the formation of the PfRh5/CyRPA complex is disrupted with PfRh5 eluted alone in the later fractions (*Figure 2C*), and CyRPA eluted as a complex with 8A7 in the earlier fractions. However, in the presence of non-inhibitory monoclonal antibody 7A6, CyRPA remains bound to PfRh5 (*Figure 2D*). The antibody 7A6 also did not bind to native CyRPA consistent with binding to a linear epitope in immunoblots (*Figure 2B*) that is not exposed on the surface of the protein. In addition, in the presence of non-inhibitory anti-PfRh5 monoclonal antibody 4G4 (*Figure 2A*), the PfRh5/CyRPA complex remained largely intact and was eluted as a tri-molecular complex in the earlier fractions, with free CyRPA eluted in the later fractions (*Figure 2E*). These results suggested that growth-inhibitory mechanism of the monoclonal antibody 8A7 is likely due to the disruption of PfRh5/CyRPA complex formation.

## The crystal structures of CyRPA alone and in complex with the Fab fragment of invasion inhibitory monoclonal antibody 8A7

We crystallized CyRPA in complex with a single 8A7 Fab fragment (see Methods). The structure of the complex was determined using diffraction data to 2.44 Å resolution by molecular replacement using a homologous Fab fragment to obtain initial phases, followed by iterative model building and crystallographic refinement (*Figure 3*). Crystals were also obtained of uncomplexed CyRPA; this structure was determined by molecular replacement, using the CyRPA component of the Fab-complexed structure as a search model, and subsequently refined using diffraction data to 3.09 Å resolution. Crystallographic data collection and refinement statistics for both structures are shown in *Table 1*.

The CyRPA structure (*Figure 3*) displays the canonical six-bladed β-propeller (6BBP) fold (*Figure 3A and B*) (*Chen et al., 2011b*). Within the β-propeller structure are four intra-sheet disulfide bonds (Cys19-Cys35, Cys90-Cys92, Cys151-Cys169 and Cys229-Cys239) and one inter-sheet disulfide bond (Cys274-Cys298) (*Figure 3A and B*). The crystal of uncomplexed CyRPA contains four copies of the protein in the crystallographic asymmetric unit, arranged with approximate 222-point group symmetry (*Figure 3C*). Within this tetramer, a pairwise association of CyRPA molecules is mediated by the formation of two eight-strand anti-parallel β-sheets, one (termed sheet '6–6') comprises two copies of the sixth β-sheet of the CyRPA monomer and the second (termed sheet '5–5') comprises two copies of the (adjacent) fifth β-sheet of the CyRPA monomer (*Figure 3C*). In each case, formation of the eight-stranded sheet is mediated by the respective outermost strands of the two constituent four-stranded sheets. These dimers of CyRPA monomers then associate with each

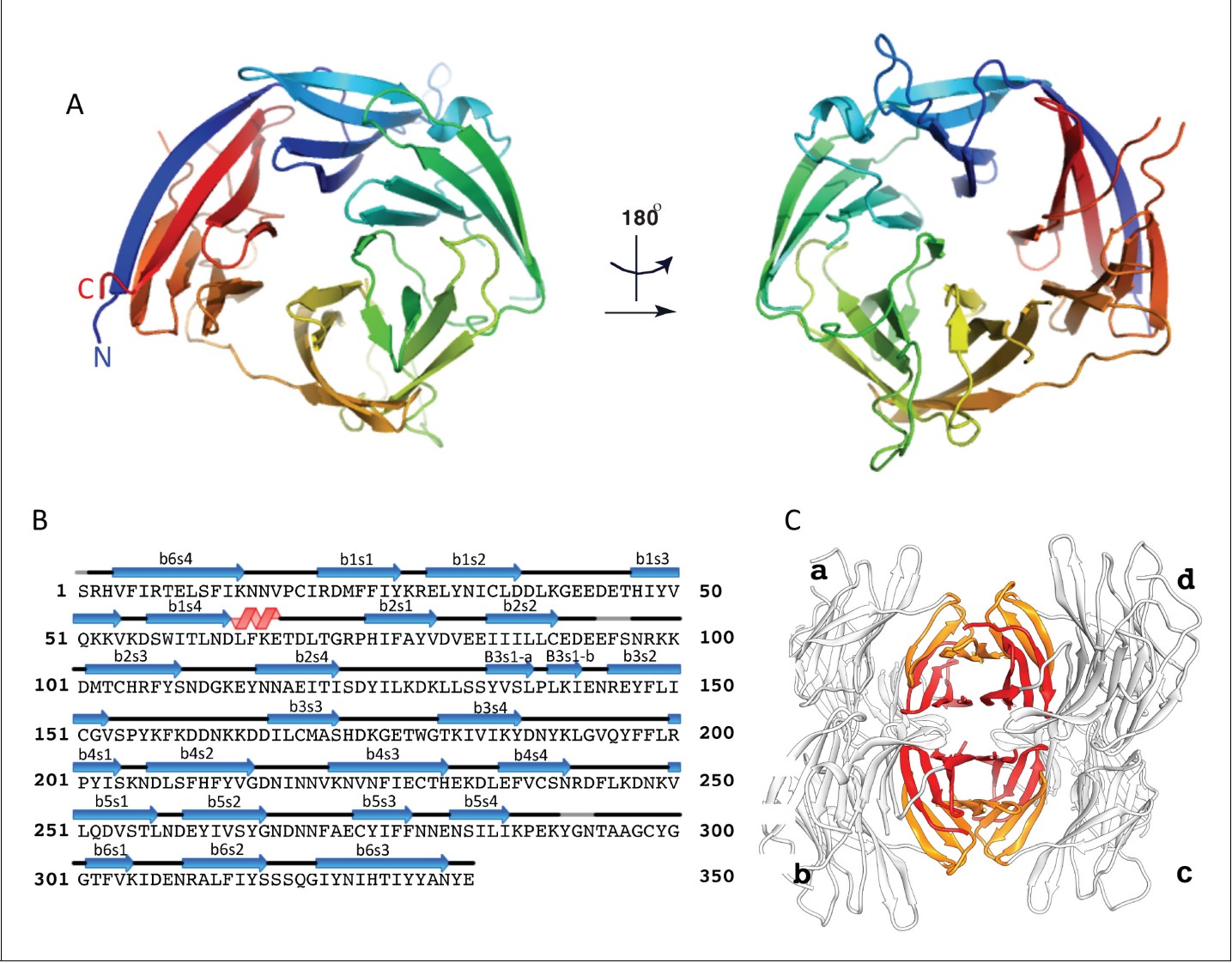

**Figure 3.** The crystal structure of the uncomplexed CyRPA. (**A**) Orthogonal view of the ribbon representation of CyRPA, colored in rainbow fashion from the N-terminus (*blue*) through to the C- terminus (*red*). (**B**) Amino acid sequence and the secondary structure of CyRPA, showing the location of the 24 canonical strands of the six-bladed β-propeller [labelled 'β*m*s*n*' where the index *n* denotes the strand and the index *m* the sheet (*n* = 1,..,4; *m* = 1,..,6)]. (**C**) Assembly of the 222 pseudo-symmetric tetramer of CyRPA within the crystallographic asymmetric unit, showing the formation of the extended 5–5 and 6–6 sheets (*orange* and *red*, respectively). Each monomer of CyRPA in the asymmetric unit is labelled a, b, c, d.

The following figure supplements are available for figure 3:

**Figure supplement 1.** Comparison of CyRPA with *C.*

**Figure supplement 2.** Recombinant CyRPA does not have sialidase activity.

other *via* the formation of a β-sandwich comprising sheet 6–6 of one CyRPA dimer and sheet 6–6 of the second CyRPA dimer (*Figure 3C*). The monomer-monomer interfaces, although extensive, are most likely an artefact of crystallization, as we find no evidence for higher-order association upon SEC of CyRPA. However, we cannot rule out such associations occurring within the hetero-complexes formed by CyRPA and its partner proteins on the surface of the merozoite.

The 6BBP fold is found in a number of enzyme families, as well as in some attachment proteins (*Chen et al., 2011b*). A DALI search (*Holm and Rosenstrom, 2010*) reveals that the closest

**Table 1.** Data collection and refinement statistics.

| | CyRPA | CyRPA/Fab |
|---|---|---|
| Wavelength (Å) | 0.954 | 0.954 |
| Resolution range (Å) | 36.45–3.09 (3.20–3.09)* | 38.34–2.44 (2.53–2.44) |
| Space group | P1 | P2₁2₁2₁ |
| Unit cell<br>  $a$, $b$, $c$ (Å)<br>  $\alpha$, $\beta$, $\gamma$ (°) | 68.7, 83.56, 95.32<br>96.76, 104.11, 115.20 | 79.95, 87.38, 145.14<br>90, 90, 90 |
| Total no. reflections | 121042 (10425) | 228716 (19565) |
| Unique no. reflections | 32250 (2919) | 38347 (3695) |
| Multiplicity | 3.8 (3.6) | 6.0 (5.3) |
| Completeness (%) | 0.98 (0.89) | 0.99 (0.97) |
| $<I/\sigma(I)>$ | 10.84 (1.73) | 14.54 (2.02) |
| Wilson $B$-factor (Å²) | 71.27 | 54.65 |
| $R_{merge}$ | 0.12 (0.86) | 0.081 (1.033) |
| $R_{meas}$ | 0.14 (1.01) | 0.089 (1.137) |
| $CC_{1/2}$ | 0.99 (0.67) | 0.998 (0.72) |
| $CC^*$ | 1.00 (0.90) | 1.00 (0.915) |
| No. reflections used in refinement | 32240 (2917) | 38345 (3695) |
| No. reflections used for $R_{free}$ | 1584 (148) | 1891 (173) |
| $R_{work}$ | 0.1871 (0.3620) | 0.1887 (0.2916) |
| $R_{free}$ | 0.2297 (0.4022) | 0.2115 (0.3000) |
| No. non-hydrogen atoms | 10627 | 6045 |
|   macromolecules | 10614 | 5990 |
|   Ligands | | 12 |
|   Protein residues | 1273 | 750 |
| RMS bonds (Å) | 0.002 | 0.003 |
| RMS angles (°) | 0.48 | 0.58 |
| Ramachandran favored (%) | 95.2 | 97.4 |
| Ramachandran allowed (%) | 4.8 | 2.6 |
| Ramachandran outliers (%) | 0 | 0 |
| Rotamer outliers (%) | 0.58 | 0.29 |
| Clash score | 6.67 | 3.39 |
| Average $B$-factor (Å²) | 82.87 | 68.03 |
|   Macromolecules | 82.91 | 68.15 |
|   Ligand | n.a. | 68.68 |
|   Solvent | 45.20 | 51.99 |
| No. TLS groups | 22 | 10 |

*Statistics for the highest-resolution shell are shown in parentheses.

structural homologue to CyRPA is the attachment glycoprotein of a henipah virus isolated from Ghanaian bats (PDB entry 4UF7 (*Lee et al., 2015*)), with the next closest homologues being various sialidases of viral, bacterial and mammalian origin (*Varghese et al., 1983*; *Newstead et al., 2008*). CyRPA has up to 16% sequence identity with all such structural homologues found by the DALI search, suggesting only distant evolutionary relationships, if any, with other thus-far structurally-characterized 6BBPs. Nevertheless, given that sialidases appear consistently as the highest-ranking

homologues, it is instructive to consider whether CyRPA contains any specific sequence motifs that are characteristic of sialidases and, indeed, whether it has any sialidase activity. We compared the sequence of CyRPA with that of *Clostridium perfringens* NanI sialidase to identify potential conserved residues that might be relevant to sialidase enzyme activity. Localization of these residues on an overlay of structures revealed the absence of any potential active site (*Figure 3—figure supplement 1*) (*Newstead et al., 2008*). There are two signature motifs for the active site of all sialidases (*Varghese et al., 1983*; *Newstead et al., 2008*). The first is a set of three conserved arginine residues that coordinate the binding of the carboxylate moiety of sialic acid (*Varghese et al., 1983*; *Newstead et al., 2008*), these residues are located respectively in the loop between the first and second β-sheets, in the loop between the third and fourth strand of the fifth β-sheet and in the loop connecting the second and third strand in the fourth β-sheet. Inspection reveals only one possible equivalent, an arginine at residue 21 in the loop between the first and second β-sheets of CyRPA (*Figure 3B*). The second signature motif is a catalytic tyrosine residue located on the loop between the fifth and sixth β-sheets (*Varghese et al., 1983*; *Newstead et al., 2008*). Again, we find no equivalent in CyRPA to this tyrosine. We further tested whether the recombinant CyRPA had sialidase activity, using the fluorogenic substrate 4-methylumbelliferyl *N*-acetyl-α-D-neuraminic acid as a substrate (see Methods), with bacterial sialidase (neuraminidase) from *Arthrobacter ureafaciens* as a positive control (*Saito et al., 1979*). No significant sialidase activity was detected (*Figure 3—figure supplement 2*). We thus conclude that CyRPA is not a functional sialidase.

### The antibody-binding region of CyRPA

Within the complex, the Fab is oriented (as defined by its long axis) approximately perpendicular to the barrel axis of the 6BBP assembly. The Fab fragment of 8A7 *Figure 3B* binds to the region of CyRPA comprising residues from the loop between the second and third strands of the first β-sheet (Gly41, Glu42 and Glu45), residues from the loop between the first and second β-sheets (Asp63, Lys66, Glu67, Thr68, Asp69, Leu70 and Thr71), residues from the loop between the second and third strands of the second β-sheet (Lys91 and Lys99), and residues comprising the N-terminal region of the fourth strand of the second β-sheet (Asn116, Asn117, Ala118 and Glu119) (*Figure 3B*, *Figure 4*). Comparison with the uncomplexed CyRPA structures reveals only minor changes in rotameric conformation and relative disposition of CyRPA residues at the interface. The antigen interaction involves residues drawn from all six complementary-determining regions (CDRs) of the Fab and buries a total of 1634 Å of the molecular surface across both molecules (*Figure 4A,B and C*). The shape complementarity of the interface ($S_c$ = 0.71) is relatively high for antibody-antigen complexes (*Lawrence and Colman, 1993*), with the surface of CyRPA epitope being predominantly negatively charged (*Figure 4C*). The interface itself involves a rich network of approximately nineteen hydrogen bonds (*Figure 4B* and *Figure 4—source data 1*).

## Discussion

The PfRh5/CyRPA/PfRipr complex plays an essential role for invasion of *P. falciparum* into human erythrocytes through binding the receptor basigin (*Volz et al., 2016*). To provide a structural basis for understanding the function of this complex and its components we have determined the structure of CyRPA in its free form. Additionally, we have determined the structure of CyRPA in complex with a Fab fragment derived from a monoclonal antibody that blocks the growth of the *P. falciparum* parasite to understand the mechanism of inhibition. CyRPA displays a canonical six-bladed β-propeller fold (*Chen et al., 2011b*), and the inhibitory monoclonal is shown to block formation of the PfRh5/CyRPA complex.

The closest structural homologue to CyRPA is the attachment glycoprotein of a henipah virus which binds to ephrinB2 and is important for host cell entry (*Lee et al., 2015*); however, it also has some similarity to bacterial and viral sialidases all of which have a six-bladed β-propeller fold (*Varghese et al., 1983*; *Crennell et al., 1996*, *1993*, *1994*). CyRPA lacks some of the important amino acid motifs required for it to be an active sialidase, and the recombinant form of the protein did not have sialidase activity suggesting it is only distantly related to these enzymes. The PfRh5/CyRPA/PfRipr complex functions together as a conditional expression of CyRPA or PfRipr and blocking of PfRh5 binding to basigin blocks merozoite invasion at the same step (*Volz et al., 2016*). Merozoites lacking PfRh5/CyRPA/PfRipr complex function can still interact and deform the erythrocyte

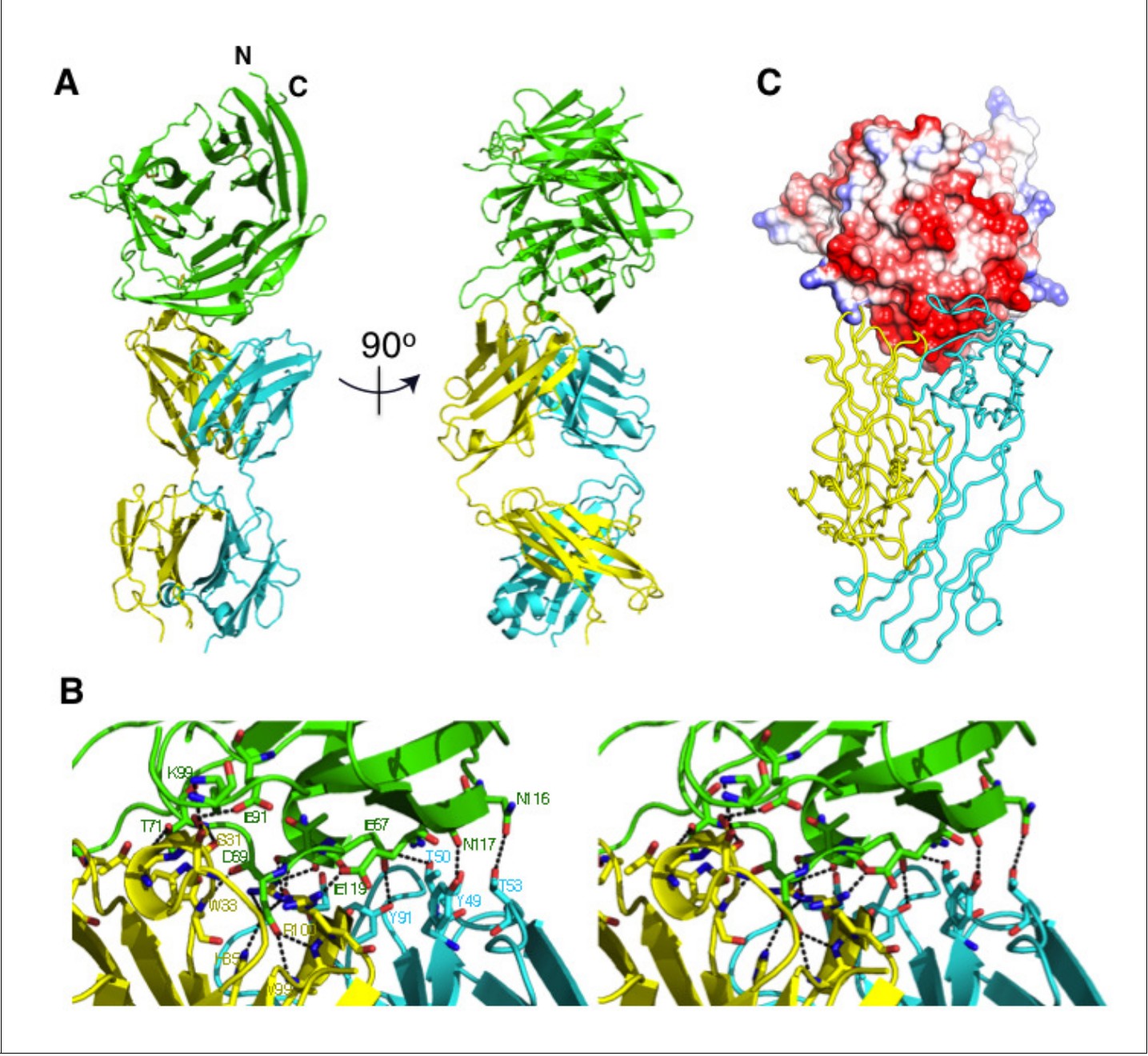

**Figure 4.** The crystal structure of the Fab 8A7 / CyRPA complex. (A) Orthogonal views of the ribbon representation of the complex (CyRPA: *green*, Fab 8A7 light chain: *cyan*; Fab 8A7 heavy chain: yellow. (B) Hydrogen bond network within the interface between Fab 8A7 and CyRPA. (C) Surface of Fab 8A7 epitope of CyRPA colored according to surface potential (*red* negative, *blue* positive), with the Fab chain colored as in (A). Surface potential was computed using CHIMERA (*Pettersen et al., 2004*).

The following source data and figure supplement are available for figure 4:

**Source data 1.** Deduced hydrogen bonds between CyRPA and Fab 8A7.

**Figure supplement 1.** Amino acid sequence of anti-CyRPA monoclonal antibody 8A7 Fab fragment for both heavy and light chains.

membrane but they cannot form the pore or discontinuity between the parasite apical end and the erythrocyte membrane through which $Ca^{2+}$ passes (*Volz et al., 2016*; *Weiss et al., 2015*). Therefore, they do not progress to form a tight junction which requires insertion of the RON complex into the host cell (*Weiss et al., 2015*).

It has been hypothesized that the PfRh5/CyRPA/PfRipr complex is directly involved in forming this pore between the merozoite and erythrocyte membrane (*Volz et al., 2016*; *Weiss et al., 2015*). The PfRh5 structure has a large coiled-coil domain that has some structural similarity with the N-terminal coiled-coil domain of SipB and other similar proteins (*Barta et al., 2012*). SipB forms part of the Salmonella type III secretion system that transports effector proteins across the host cell membrane. Additionally, there are six-bladed $\beta$–propeller structured proteins, similar to CyRPA, that play a role in the formation of protein translocation channels by bacterial pathogens to inject virulence factors (*Meusch et al., 2014*). Whilst these similarities are tantalizing, the link between the formation of a pore between the merozoite surface and the erythrocyte membrane by the PfRh5/CyRPA/PfRipr is circumstantial and requires further evidence (*Volz et al., 2016*; *Weiss et al., 2015*).

Our experiments show that CyRPA binds to PfRh5 in the PfRh5/CyRPA/PfRipr tripartite complex and it is likely that it also interacts directly with PfRipr (*Volz et al., 2016*). The subcellular localization of CyRPA and PfRipr is in the micronemes whereas PfRh5 is at the neck of the rhoptries and the tripartite complex only forms at the interface between the apical end of the merozoite and the erythrocyte membrane. All three proteins are spread over the free merozoite surface but generally CyRPA and PfRipr co-localise whereas PfRh5 does not. This suggests that the PfRh5/CyRPA/PfRipr complex forms during the invasion process when it is required and that its assembly is finely regulated, most likely requiring the function of other unknown proteins. It is also not understood how the PfRh5/CyRPA/PfRipr complex or CyRPA associates with the merozoite membrane as none of these proteins have a transmembrane domain or a GPI anchor (*Volz et al., 2016*).

The anti-CyRPA monoclonal antibody 8A7 directly abrogates the interaction of CyRPA with PfRh5 and as a result directly blocks merozoite invasion of *P. falciparum* into human erythrocytes. This suggests that the PfRh5 binding site for CyRPA overlaps with (or lies close to) the antibody conformational epitope, defined by residues within the loop between the second and third strands of the first $\beta$-sheet, the loop between the first and second $\beta$-sheets, and residues at the N-terminus region of the fourth strand of the second $\beta$-sheet. The absence of large conformational change within CyRPA upon Fab complexation rules out an allosteric mechanism of abrogating a complex formation with PfRh5. Therefore, blocking of the CyRPA and PfRh5 interaction by antibodies is consistent with a functional PfRh5/CyRPA/PfRipr complex assembling at the interface between the invading merozoite and erythrocyte membrane (*Volz et al., 2016*).

CyRPA is a highly conserved protein in *P. falciparum* and the antibody binding epitope as well as the remainder of the protein is not under selective pressure. As a result, it is a promising vaccine candidate as antibodies inhibit merozoite invasion in vitro as well as in mice engrafted with human erythrocytes (*Reddy et al., 2014*; *Dreyer et al., 2012*). Our results elucidate the structural basis for the binding and inhibition of antibodies to CyRPA that block merozoite invasion. This information will facilitate the development of this candidate in potential vaccines most likely in a combination vaccine with PfRh5 and PfRipr or in the complex form of PfRh5/CyRPA/PfRipr against malaria.

## Materials and methods

### Protein expression and purification

*Plasmodium falciparum* (3D7) full-length mature CyRPA with the last residues of the three potential glycosylation site motifs mutated to alanine ([116]NNS to NNA, [293]NTT to NTA and [299]NRT to NRA) was cloned in to the vector pgpHFT (*Xu et al., 2010*) and expressed in insect cells as described (*Chen et al., 2011a*), but employing in this instance Sf21 cells rather than Hi5 cells. The supernatant containing the secreted recombinant protein was harvested, centrifuged and passed over an anti-FLAG M2 agarose (Sigma) column. After extensive washing, bound proteins were eluted from the column with FLAG peptide at a concentration of 100 µg/ml, concentrated and then further purified by size-exclusion chromatography (SEC) with a Superdex 200 column (GL 10/300, GE Healthcare) in 25 mM HEPES, 150 mM NaCl, pH 7.2. For crystallization of CyRPA, the tandem hexa-His and FLAG tags were removed by digestion with a TEV protease and purified protein was recovered by Ni-resin

and/or SEC. The Fab fragment of anti-CyRPA monoclonal 8A7 (see below for details of monoclonal antibody production and growth inhibition assays) was prepared by digesting the monoclonal antibody parent with papain and then purifying by SEC using a Superdex 200 column (GL 10/300; GE Healthcare). The CyRPA/Fab complex was prepared by combining an excess of CyRPA with the Fab fragment followed by purification of the complex by SEC.

To obtain the amino acid sequence of the VH and VL segments of 8A7 total RNA was isolated from the hybridoma cells following the technical manual of TRIzol Reagent. Total RNA was then reverse transcribed into cDNA using IgG1 kappa-specific anti-sense primers or universal primers following the technical manual of PrimeScriptTM 1st Strand cDNA Synthesis Kit. The antibody fragments of VH and VL were amplified according to the standard operating procedure of rapid amplification of cDNA ends (RACE) of GenScript (New Jersey). Amplified antibody fragments were cloned separately into a standard cloning vector separately. Colony PCR was performed to screen for clones with inserts of correct sizes. No fewer than five colonies with inserts of correct sizes were sequenced for each fragment. The sequences of different clones were aligned and the consensus sequence of these clones was obtained.

## Production of monoclonal antibodies and growth inhibition assays

All procedures with mice were approved by the Walter and Eliza Hall Medical Research Institute Animal Ethics Committee. BALB/c mice were immunized with 20 µg of recombinant CyRPA in Freund's complete adjuvant (1st injection; Sigma Aldrich) followed by two subsequent injections of 20 µg CyRPA in Freund's incomplete adjuvant (two subsequent injections; Sigma Aldrich, St. Louis, MO) then 40 µg CyRPA in saline and 50 µg CyRPA in saline. Mice were euthanized and spleens harvested using aseptic technique. Spleen cells were fused with myeloma cells. Anti-CyRPA antibody-producing hybridoma cell lines were identified 14 days after fusion, using ELISA with recombinant CyRPA. Positive hybridomas were sub-cloned to obtain monoclonal cell lines. Positive antibodies were subsequently screened by immuno-blot with mature schizonts. Monoclonal antibodies 7A6, 8B9, 3D1, 8A7 and 5B12 were selected for further study and tested to determine if they inhibit *P. falciparum* growth in vitro. 8A7 was used for co-crystallography studies and is isotype IgG1 kappa. Non-inhibitory anti-PfRh5 monoclonal antibody 4G4 and inhibitory anti-PfRipr monoclonal antibody 5G6 were included as controls.

Two cycle growth inhibition assays were performed, using 3D7 strain of *P. falciparum*, as described to determine if the monoclonal antibodies inhibited parasite growth (*Healer et al., 2013*). Serial dilutions of IgG in PBS, starting at 2 mg/ml were added to *P. falciparum*-infected erythrocytes (3D7) at a parasitaemia of 0.1%. Parasitaemia was counted after 96 hr by flow cytometry and specific growth inhibition calculated relative to parasites grown in non-immune IgG. *P. falciparum* strain 3D7 has been confirmed by whole genome sequencing and was originally obtained from Prof. David Walliker, Edinburgh University.

## Immuno-blot screen of anti-CyRPA monoclonal antibodies

Synchronized schizont stage 3D7 parasites that expressed HA-tagged CyRPA (*Volz et al., 2016*) were harvested and erythrocytes lysed using 0.15% saponin. Proteins were solubilized from the saponin pellet with reducing and non-reducing SDS-PAGE sample buffer and separated on 4–12% *bis*-Tris precast NuPAGE gel (Life Technologies, Carlsbad, Ca, USA) then transferred to a nitrocellulose membrane. The membranes were probed with the anti-CyRPA monoclonal antibodies (1/1000) and anti-HA mouse monoclonal (1/1000) as a positive control and proteins detected by enhanced chemiluminescence (ECL, GE Healthcare).

## SEC and co-immunoprecipitation

SEC was performed on an analytical Superdex 200 column or Superose-6 column (24 ml, GE Healthcare) and proteins were eluted with 25 mM HEPES, 150 mM NaCl, pH 7.2 or 25 mM Tris, 150 mM NaCl, pH 8.5. Co-immunoprecipitation of CyRPA with FLAG-tagged PfRh5 was performed using anti-FLAG M2 agarose (Sigma). Briefly, 5 µg FLAG-tagged PfRh5 and 5 µg CyRPA were incubated with 20 µl anti-FLAG M2 agarose beads for 1 hr on ice. After extensive washing with 25 mM Tris, pH 8.5 buffer containing 150 mM NaCl, the bound proteins were eluted with the FLAG peptide at a

concentration of 100 µg/ml in the same Tris buffer, analysed by SDS-PAGE and stained with Coomassie Blue R250.

## Crystallization and structure determination

Crystallization trials for both the Fab-bound and the uncomplexed CyRPA protein were initially performed in sitting-drop, 96-well format at 18°C performed at the CSIRO Collaborative Crystallization Centre (Parkville, Australia), followed by in-house refinement. Diffracting crystals of CyRPA were finally obtained from hanging drops containing 9% PEG1000, 9% PEG8000, 0.25 M potassium iodide and 50 mM potassium thiocyanate. Diffracting crystals of CyRPA/Fab complex were finally obtained from hanging drops containing 11% PEG5000 monomethyl ether and 0.2M sodium acetate, pH 5.5. For data collection, crystals were transferred to cryo-protection solutions (12% PEG1000, 12% PEG8000, 0.25 M potassium iodide, 50 mM potassium thiocyanate and 25% ethylene glycol for CyRPA crystals; 15% PEG5000 monomethyl ether and 0.2M sodium acetate, pH 5.5% and 25% ethylene glycol for CyRPA/Fab complex crystals) and cryo-cooled by plunging into liquid nitrogen. X-ray diffraction data were collected on beamline MX2 at the Australian Synchrotron at a temperature of ~100 K (*McPhillips et al., 2002*).

Diffraction data were processed and scaled with XDS (*Kabsch, 2010*). Resolution limits were set to 3.09 Å for the native CyRPA crystal data and to 2.44 Å for the CyRPA/Fab complex crystal data. For the structure solution of CyRPA/Fab complex, initial phases were obtained by molecular replacement using PHASER (*McCoy et al., 2007*) with the structure of a murine Fab fragment (PDB code: 5L9D; unpublished) as a search model within PHASER (*Kabsch, 2010*). Model building and refinement then continued within PHENIX (*Adams et al., 2010*), with automated building and morphing routines leading to a model comprising the majority of the CyRPA/Fab sequence. Further rounds of refinement and manual rebuilding were undertaken using PHENIX (*Adams et al., 2010*) and COOT (*Emsley et al., 2010*). The structure of the native CyRPA was solved by molecular replacement with coordinates of CyRPA—derived from the refined CyRPA/Fab structure—used as a search model. Data processing and refinement statistics for both structures are provided in *Table 1*.

## Sialidase assay

Sialidase (i.e., neuraminidase) activity was assayed using the fluorogenic substrate 4-methylumbelliferyl *N*-acetyl-$\alpha$-D-neuraminic acid sodium salt (4 mU-NeuNAc). Reaction volumes (100 µl) containing 4 mU-NeuNAc (100 µM) at pH 5.0 (50 mM citrate, 150 mM NaCl), pH 6.5 (50 mM MES, 150 mM NaCl) and pH 8.0 (50 mM Tris-HCl, 150 mM NaCl) were incubated with BSA (5 µg/ml), CyRPA (5.0 µg/ml) or *Arthrobacter ureafaciens* sialidase (neuraminidase) (Sigma-Aldrich N3786, 0.1 µg/ml) for 4 hr at 37°C. Each reaction was quenched with glycine buffer (900 µl, 1 M, pH 10) and fluorescence measured ($\lambda_{ex}$ = 365 nm, $\lambda_{em}$ = 445 nm) on a Cary Eclipse fluorimeter (Agilent Technologies). These values, obtained in triplicate, were normalized with respect to the fluorescence measured for 10 µM 4-methylumbelliferone in glycine buffer (0.9 M, pH 10) to determine the completeness of each reaction.

## Acknowledgements

We thank the Victorian Red Cross Blood Bank for the supply of blood and Kaye Wycherley and Paul Masendycz of the WEHI Monoclonal Facility for production of monoclonal antibodies. Part of the research was undertaken on the MX2 beamline at the Australian Synchrotron, Victoria, Australia.

## Additional information

### Funding

| Funder | Grant reference number | Author |
|---|---|---|
| Victorian State Government | Independent Research Institute Infrastructure Support Scheme | Lin Chen<br>Yibin Xu<br>Wilson Wong<br>Jennifer K Thompson<br>Ethan D Goddard-Borger<br>Michael C Lawrence |

| | | Alan F Cowman |
|---|---|---|
| National Health and Medical Research Council | Independent Research Institute Infrastructure Support Scheme | Lin Chen<br>Yibin Xu<br>Wilson Wong<br>Jennifer K Thompson<br>Ethan D Goddard-Borger |
| Howard Hughes Medical Institute | 55007645 | Alan F Cowman |
| National Health and Medical Research Council | 637406 | Alan F Cowman |
| Path/Malaria Vaccine Initiative | 07608-COL | Alan F Cowman |
| United States Agency for International Development | 07608-COL | Alan F Cowman |

The funders had no role in study design, data collection and interpretation, or the decision to submit the work for publication.

### Author contributions

LC, Conception and design, Acquisition of data, Analysis and interpretation of data, Drafting and revising the article; YX, WW, JKT, JH, EDG-B, Acquisition of data, Analysis and interpretation of data; MCL, Analysis and interpretation of data, Drafting and revision of manuscript; AFC, Conception and design, Analysis and interpretation of data, Drafting and revision of the manuscript

### Author ORCIDs

Alan F Cowman, http://orcid.org/0000-0001-5145-9004

## Additional files

### Major datasets

The following datasets were generated:

| Author(s) | Year | Dataset title | Dataset URL | Database, license, and accessibility information |
|---|---|---|---|---|
| Lin Chen, Yibin Xu, Wilson Wong, Jennifer K Thompson, Julie Healer, Ethan D Goddard-Borger, Michael C Lawrence, Alan F Cowman | 2017 | Structural basis for inhibition of erythrocyte invasion by antibodies to *Plasmodium falciparum* protein CyRPA | http://www.rcsb.org/pdb/explore/explore.do?structureId=5TIH | Publicly available at the RCSB Protein Data Bank (Accession no: 5TIH) |
| Lin Chen, Yibin Xu, Wilson Wong, Jennifer K Thompson, Julie Healer, Ethan D Goddard-Borger, Michael C Lawrence, Alan F Cowman | 2017 | Structural basis for inhibition of erythrocyte invasion by antibodies to *Plasmodium falciparum* protein CyRPA | http://www.rcsb.org/pdb/explore/explore.do?structureId=5TIK | Publicly available at the RCSB Protein Data Bank (Accession no: 5TIK) |

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
