## [Decision Letter]

Thank you for submitting your article "Structural basis for inhibition of erythrocyte invasion by antibodies to *Plasmodium falciparum* protein CyRPA" for consideration by *eLife*. Your article has been reviewed by three peer reviewers, one of whom Stephen C. Harrison (Reviewer #1), is a member of our Board of Reviewing Editors, and the evaluation has been overseen by John Kuriyan as the Senior Editor.

The reviewers have discussed the reviews with one another and the Reviewing Editor has drafted this decision to help you prepare a revised submission.

Summary:

Chen at all investigate the structure and inhibition of PfCyRPA, an essential protein for the invasion of erythrocytes by *Plasmodium falciparum*. They show that CyRPA interacts directly with RH5. They identify monoclonal antibodies with growth inhibitory activity and present data suggesting that one of these prevents CyRPA from interacting with Rh5. They then present the structure of CyRPA, both alone and in complex with one of these antibodies. This is an interesting piece of work that analyzes an important protein and putative vaccine candidate.

Major points needing attention/revision:

1) The structural biology is generally convincing, but the data cut-off used is too generous. The values for I/sig(I) and CC_1/2_ for the outer shell are extremely low. In addition, the overall *R_merge_*is very high, and the *R_merge_* in the outer shell is exceptionally high. The structure of the complex is actually the better of the two; the claimed resolution is 2.26 Å, but even 2.5Å (a more conservative cutoff) is accurate enough for the conclusions the authors draw. The data processing and refinement need to be repeated using more appropriate resolution cutoffs, and the new, lower, resolution should be reported in the main text.

2) In the parasite inhibition studies, a control antibody should be included for comparison. For a reader to judge how effective these CyRPA antibodies are at inhibiting invasion, it would also be valuable to include one or more other inhibitory antibodies (i.e. those that bind to PfRH5) included in this analysis so that the study would go beyond a demonstration of some inhibitory potential and allow the reader to see how inhibitory potency of this mAb compares with those of mAbs targeting other leading vaccine candidates. The field generally lacks this comparative analysis, which makes deciding which vaccine candidates to take forward challenging.

3) The authors show data to suggest 8A7 directly prevents RH5 binding to CyRPA, while 7A6 does not (Figure 2). While this conclusion is likely, the data presented are not as convincing as they should be. (a) It appears that the binding of RH5 and CyRPA is weak as indicated by the pulldown experiments (Figure 1) and the coelution profile (1B). Given the weak affinity it is plausible that the concentrations of RH5 and CyRPA used in the inhibition experiment (2C) might not have allowed complex formation, leading to an inaccurate conclusion that the inhibitory effect of 8A7 comes from blocking the interaction. (b) The authors use 7A6 as a control to demonstrate this antibody does not disrupt the complex and does not prevent parasite growth. Because 7A6 does not bind folded CyRPA and therefore cannot possibly disrupt binding, it is not a suitable control.

3) The authors state that they find no evidence for higher-order association from SEC of CyRPA. Figure 1 and Figure 2 seem to indicate otherwise. RH5 has a larger mass than CyRPA but elutes later than CyRPA. Thus, CyRPA appears to elute far earlier than a monomer would, and one might conclude that it is an oligomer. Additional data should be provided to examine the oligomeric state in solution. Oligomerization is important for complex formation of a number of parasite proteins (reviewed in Paing, et al. PLOS Pathogens (2014) 10(6): e1004120). It might be prudent to cite this and other related work. Finally, how does the 8A7 epitope correlate with the putative tetrameric contacts in CyRPA?

4) It would be good to have a direct measurement, for example by SPR, of the CyRPA:RH5 interaction, in the presence and absence of the antibody. The relative affinities and on- and off- rates for these interactions will probe how this antibody-mediated inhibition works and show whether the kinetics of the two binding events are compatible with the disruption of a pre-formed CyRPA:CyRPA complex.

5) Please provide a DNA sequence for the entire Fab (e.g., in a supplemental figure), as it is important for assessing the structure. (Otherwise, a reader will need to consult the deposited PDB file, which can be awkward.)

---

## [Author Response]

Major points needing attention/revision:

1) The structural biology is generally convincing, but the data cut-off used is too generous. The values for I/sig(I) and CC_1/2_ for the outer shell are extremely low. In addition, the overall R_merge_ is very high, and the R_merge_ in the outer shell is exceptionally high. The structure of the complex is actually the better of the two; the claimed resolution is 2.26 Å, but even 2.5Å (a more conservative cutoff) is accurate enough for the conclusions the authors draw. The data processing and refinement need to be repeated using more appropriate resolution cutoffs, and the new, lower, resolution should be reported in the main text.

We have reprocessed the data, refined the structure and modified Table 1 as requested. CyRPA is now a resolution of 3.09 Å. CyRPA/Fab complex is now 2.44 Å. These are reported in the main text.

2) In the parasite inhibition studies, a control antibody should be included for comparison. For a reader to judge how effective these CyRPA antibodies are at inhibiting invasion, it would also be valuable to include one or more other inhibitory antibodies (i.e. those that bind to PfRH5) included in this analysis so that the study would go beyond a demonstration of some inhibitory potential and allow the reader to see how inhibitory potency of this mAb compares with those of mAbs targeting other leading vaccine candidates. The field generally lacks this comparative analysis, which makes deciding which vaccine candidates to take forward challenging.

We have now included an inhibitory anti-Ripr monoclonal antibody 5G6 as a positive control and a non-inhibitory anti-Rh5 monoclonal antibody 4G4 as a negative control.

3) The authors show data to suggest 8A7 directly prevents RH5 binding to CyRPA, while 7A6 does not (Figure 2). While this conclusion is likely, the data presented are not as convincing as they should be. (a) It appears that the binding of RH5 and CyRPA is weak as indicated by the pulldown experiments (Figure 1) and the coelution profile (1B). Given the weak affinity it is plausible that the concentrations of RH5 and CyRPA used in the inhibition experiment (2C) might not have allowed complex formation, leading to an inaccurate conclusion that the inhibitory effect of 8A7 comes from blocking the interaction. (b) The authors use 7A6 as a control to demonstrate this antibody does not disrupt the complex and does not prevent parasite growth. Because 7A6 does not bind folded CyRPA and therefore cannot possibly disrupt binding, it is not a suitable control.

The concentration of PfRh5 and CyRPA used in Figure 2 are similar to those used in Figure 1 and therefore Rh5/CyRPA complex should form if anti-CyRPA 8A7 doesn’t block the interaction, as in the case of Figure 2. Thus we are confident in the conclusion we have made.

For panel 2D, we agree that anti-CyRPA antibody 7A6 that doesn’t bind to folded CyRPA is not the perfect control. Nevertheless, we have demonstrated that without a blocking antibody, Rh5 would form a complex with CyRPA.

However, to address the reviewers comments we have screened a larger number of monoclonals to both PfRh5 and CyRPA. All of the CyRPA non-inhibitory monoclonals produced react to internal epitopes as is the case for 7A6 so were not useful as extra controls required by the reviewers. We have retained the experiment using 7A6 as a control as we believe it is relevant. However, we have identified a monoclonal antibody to PfRh5 (Rh5 4G4), which is non-inhibitory, and this binds to the PfRh5/CyRPA complex but does not disrupt it. This adds an additional control requested in Figure 2 panel D.

3) The authors state that they find no evidence for higher-order association from SEC of CyRPA. Figure 1 and Figure 2 seem to indicate otherwise. RH5 has a larger mass than CyRPA but elutes later than CyRPA. Thus, CyRPA appears to elute far earlier than a monomer would, and one might conclude that it is an oligomer. Additional data should be provided to examine the oligomeric state in solution. Oligomerization is important for complex formation of a number of parasite proteins (reviewed in Paing, et al. PLOS Pathogens (2014) 10(6): e1004120). It might be prudent to cite this and other related work. Finally, how does the 8A7 epitope correlate with the putative tetrameric contacts in CyRPA?

The CyRPA we used in the experiments is monomeric based on SEC analyses. In Figure 1, CyRPA formed a complex with Rh5 and therefore was eluted in the earlier fractions compared to free CyRPA. In Figure 2, antibody 8A7 bound to CyRPA, disrupting the Rh5/CyRPA complex, and CyRPA observed in the earlier fractions is in complex with the antibody.

Thus, in Figure 1 and Figure 2, CyRPA doesn’t elute earlier than Rh5 and in fact CyRPA is indeed eluted slightly later than Rh5 as presented in Figure 2.

We have altered 1B and 2C figure legends to more fully explain this.

The antibody 8A7 epitope lies outside the interface of the tetrameric CyRPA in the crystallographic asymmetric unit.

4) It would be good to have a direct measurement, for example by SPR, of the CyRPA:RH5 interaction, in the presence and absence of the antibody. The relative affinities and on- and off- rates for these interactions will probe how this antibody-mediated inhibition works and show whether the kinetics of the two binding events are compatible with the disruption of a pre-formed CyRPA:CyRPA complex.

We agree that SPR would be very useful to have and before submitting the initial manuscript we tried to get such data with which we were confident. However, using SPR it can be very difficult to study protein-protein interaction (apart from antibody-antigen interaction). As the reviewers would know the most difficult part is regeneration of the chip to allow multiple injections to derive kinetic data. Many proteins don’t survive the regeneration conditions and this was our major problem. Furthermore, it is very difficult to control how the protein is coupled to a chip and to find a valid control. As said above we attempted SPR to obtain the data requested by the reviewer but were unable to generate reliable data that we were comfortable in publishing.

Despite this we believe that our SEC data shows that PfR5 and CyRPA form a stable complex. As stated above the CyRPA we have generated appears to be a monomer.

5) Please provide a DNA sequence for the entire Fab (e.g., in a supplemental figure), as it is important for assessing the structure. (Otherwise, a reader will need to consult the deposited PDB file, which can be awkward.)

We have included a supplemental figure for the sequence of the entire Fab fragment.